# Falcon: Fast Spectral Inference on Encrypted Data

Qian Lou
Indiana University Bloomington
louqian@iu.edu

Wen-jie Lu
Alibaba Group
juhou.lwj@alibaba-inc.com

Cheng Hong
Alibaba Group
vince.hc@alibaba-inc.com

Lei Jiang
Indiana University Bloomington
jiang60@iu.edu

## Abstract

Homomorphic Encryption (HE) based secure Neural Networks(NNs) inference is one of the most promising security solutions to emerging Machine Learning as a Service (MLaaS). In the HE-based MLaaS setting, a client encrypts the sensitive data, and uploads the encrypted data to the server that directly processes the encrypted data without decryption, and returns the encrypted result to the client. The client'S data privacy is preserved since only the client has the private key. Existing HE-enabled Neural Networks (HENNs), however, suffer from heavy computational overheads. The state-of-the-art HENNs adopt ciphertext packing techniques to reduce homomorphic multiplications by packing multiple messages into one single ciphertext. Nevertheless, rotations are required in these HENNs to implement the sum of the elements within the same ciphertext. We observed that HENNs have to pay significant computing overhead on rotations, and each of rotations is $\sim 10\times$ more expensive than homomorphic multiplications between ciphertext and plaintext. So the massive rotations have become a primary obstacle of efficient HENNs.

In this paper, we propose a fast, frequency-domain deep neural network called Falcon, for fast inferences on encrypted data. Falcon includes a fast Homomorphic Discrete Fourier Transform (HDFT) using block-circulant matrices to homomorphically support spectral operations. We also propose several efficient methods to reduce inference latency, including Homomorphic Spectral Convolution and Homomorphic Spectral Fully Connected operations by combining the batched HE and block-circulant matrices. Our experimental results show Falcon achieves the state-of-the-art inference accuracy and reduces the inference latency by $45.45\% \sim 85.34\%$ over prior HENNs on MNIST and CIFAR-10.

## 1   Introduction

Homomorphic Encryption (HE)-enabled neural networks (NNs) [1, 2, 3] are designed for secure Machine Learning as a Service (MLaaS). In HE-enabled MLaaS, a client encrypts his/her data and uploads the encrypted data to a server in the cloud. The server computes inferences on the encrypted data and returns the encrypted output to the client. The server cannot decrypt the encrypted input or output during an inference. However, HE naturally supports only linear layers of a neural network. Some *interactive* HE-enabled NNs (HENNs) [4, 5, 6] take advantage of multi-party computation (MPC) to get the client involved in the computation of activation layers by exchanging several gigabyte data with the client during an inference, while other *non-interactive* HENNs [1, 2, 3] approximate activations by a square function to perform secure inferences without involving the

client. A non-interactive HE-enabled NN is a practical MLaaS solution with competitive accuracy for particular clients who have limited computing power and small network bandwidth.

However, both interactive and non-interactive HE-enabled inferences are slow. An inference of state-of-the-art HENNs [5, 3] on an encrypted CIFAR-10 image costs several hundred seconds. Their long inference latency is caused by expensive HE rotations. Modern HE cryptosystems, e.g., BFV [7], pack a vector consisting of small integers into a single large integer, so that they can allow concurrent HE arithmetic operations to happen on individual integers by performing a single operation on the large integer. The single instruction multiple data (SIMD) computing style of HE significantly reduces inference latency of HENNs from multiple hours to several hundred seconds. However, each accumulation in linear layers of a HE-enabled NN requires a rotation operation to shuffle small integers packed into a large integer. As a result, rotations consume $> 90\%$ of inference latency of a HE-enabled NN on an encrypted CIFAR-10 image.

Recent interactive HENNs [4, 8] use frequency-domain convolutions [9, 10] to perform only element-wise multiplications in their linear layers to eliminate expensive HE rotations. A plaintext image of a client is first converted to its frequency-domain representation by discrete Fourier transform (DFT), encrypted to a ciphertext, and then sent to a cloud server. Instead of HE multiply-accumulate (MAC) operations, only HE element-wise multiplications are required to perform frequency-domain convolutions on the server. After receiving the encrypted frequency-domain linear layer output, the client decrypts it and converts the plaintext frequency-domain output to a normal linear layer output by inverse DFT (IDFT). At last, the client performs activations on the normal linear layer output, and then moves to the next layer. The frequency-domain convolutions greatly reduce inference latency of interactive HENNs by $30\% \sim 40\%$.

However, naïvely using frequency-domain convolutions in non-interactive HENNs prolongs inference latency. Each HE operation introduces a certain amount of noise into the encrypted data. When the accumulated noise in the encrypted data is larger than the noise budget of a HENN, the encrypted data cannot be correctly decrypted. Therefore, the total number of HE operations along the critical path of a HENN decides the noise budget. A larger noise budget increases the latency of each HE operation. In interactive HENNs, the server sends the output to the client at the end of each linear layer. The client has to participate the computation of each activation layer. The client performs DFT and IDFT on plaintext data, and thus does not increase the number of HE operations at all. On the contrary, non-interactive HENNs homomorphically compute all linear and activation layers on the server without involving client. DFT and IDFT applied in non-interactive HENNs happen on the encrypted data, and thus should be homomorphic. Homomorphic DFT and IDFT greatly increase the noise budget of a non-interactive HENN by adding more HE operations to its critical path. Based on our estimation, the enlarged noise budget significantly prolongs inference latency of a frequency-domain non-interactive HENN by $> 100\%$.

In this paper, we propose *Falcon* for fast non-interactive privacy-preserving inference. Our contributions can be summarized as follows.

- We propose a novel HE DFT algorithm to homomorphically and efficiently convert an encrypted input to its encrypted frequency-domain representation.

- We propose a fast HE-enable convolution technique and a fully-connected technique on spectral domain using block circulant weight matrices.

- We consider the improvements and overhead of proposed techniques on both HE noise growth and HE parameters selection. Our experiments prove that Falcon reduces the inference latency by $45.45\% \sim 85.34\%$ over prior HENNs on MNIST and CIFAR-10.

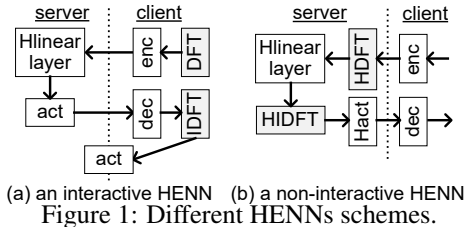

(a) an interactive HENN    (b) a non-interactive HENN
Figure 1: Different HENNs schemes.

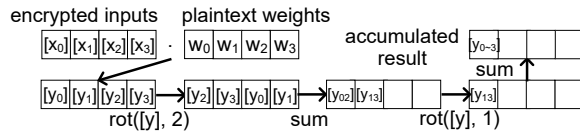

Figure 2: A homomorphic dot-product.

## 2 Background

### 2.1 Secure Neural Network Inference

Recent works [1, 2, 3, 4, 5, 6] use HE to implement linear layers of a HENN for MLaaS. However, HE cannot support non-linear activation layers. As Figure 1(a) shows, interactive HENNs [4, 5, 6] take advantage of MPC and secrete sharing to make the server to send the output to the client at the end of each linear layer, and get the client involved in the computation of each activation layer. In contrast, non-interactive HENNs [1, 2, 3] approximate activations by a square function to compute an entire secure inference without involving the client, as shown in Figure 1(b). Compared to interactive HENNs, non-interactive HENNs have a lower requirement on the computing power and network bandwidth of the client, thereby becoming more friendly to low-power mobile devices. A state-of-the-art interactive HENN, Delphi [5], has to exchange 2GB data between the client and server for only a ResNet-32 inference on an encrypted CIFAR-10 image. In this paper, we focus on accelerating non-interactive HENNs. Particularly, we select LoLa [3] implemented by BFV [7] as our baseline, due to its state-of-the-art inference accuracy and latency. Compared to other HE cryptosystems such as CKKS [11], the BFV-based LoLa improves inference latency by 30%.

### 2.2 Homomorphic Encryption

*Homomorphic Encryption.* HE allows operations on encrypted data without requiring access to the secret key [7]. Given a public key $pk$, a private key $sk$, an encryption function $\epsilon()$, and a decryption function $\sigma()$, a HE operation $\otimes$ can be defined if there is another operation $\times$ such that $\sigma(\epsilon(x_1, pk) \otimes \epsilon(x_2, pk), sk) = \sigma(\epsilon(x_1 \times x_2, pk), sk)$, where $x_1$ and $x_2$ are plaintexts. Each HE operation introduces a certain amount of noise into encrypted data. When the accumulated noise is larger than a noise budget, errors happen during HE decryption. A *bootstrapping* operation [12] is extremely expensive, although it can reduce the noise in encrypted data. Prior HENNs use *leveled* HE defining a noise budget to compute only a limited number of HE operations.

*SIMD and Rotation.* Modern HE cryptosystems, e.g., BFV [7], support single instruction multiple data (SIMD) vectors by encoding multiple integers into a larger integer based on Chinese Remainder Theorem. For instance, as Figure 2 shows, an encrypted input integer vector $[x_0, x_1, x_2, x_3]$ can be encrypted into $m_x$, while another weight integer vector $[w_0, w_1, w_2, w_3]$ can be encrypted into $m_w$. By computing a HE multiplication between $m_x$ and $m_w$, four HE multiplications are simultaneously performed on individual integers, i.e., $[x_0 \cdot w_0, x_1 \cdot w_1, x_2 \cdot w_2, x_3 \cdot w_3] = [y_0, y_1, y_2, y_3]$. A HE cryptosystem also supports rotations to shuffle individual integers in a packed vector. For instance, rotating the vector $[y_0, y_1, y_2, y_3]$ by 2 results in the vector $[y_2, y_3, y_0, y_1]$. A HE rotation is computationally expensive [4] and introduces non-trivial noise into the encrypted data.

*Homomorphic Multiply-Accumulate.* The major operation in linear layers of a HENN is homomorphic MAC, as shown in Figure 2. For a fully-connected (FC) layer, we assume the encrypted input vector $\mathbf{x}$ includes $n$ elements, the encrypted output vector $\mathbf{y}$ has $m$ elements, and the plaintext weight matrix $\mathbf{w}$ has a dimension of $n \times m$. For instance, for each row of $\mathbf{w_j}$ ($0 \leq j \leq m - 1$) and $[\mathbf{x}]$, we have $[y_j] = [\mathbf{w_j} \cdot \mathbf{x}]$ is computed as

$$[\mathbf{y_j}] = MulPC(\mathbf{w_j}, [\mathbf{x}]), [y_j] = \sum_{i=1}^{log_2 n} rot([\mathbf{y}], \frac{n}{2^i}) \tag{1}$$

where $MulPC$ indicates a HE SIMD multiplication between a packed plaintext and a packed ciphertext; and $rot$ means a HE rotation. As Equation 1 describes, $log_2 n$ HE rotations are required to accumulate an element of $\mathbf{y}$. We summarized the HE operation number and noise of a FC layer of LoLa in Table 1. Among all HE operations, rotations dominate inference latency of LoLa. In this paper, we focus on eliminating HE rotate-and-accumulate operations.

### 2.3 Frequency-Domain Convolution

*Interactive HENNs.* Prior interactive HENNs [4, 8] use frequency-domain convolutions [9, 10] to perform only element-wise multiplications in their HE linear layers to reduce rotation overhead, as shown in Figure 1(a). Based on Convolution Theorem, the convolutions in space domain are equivalent to element-wise products in frequency domain. Therefore, we have

$$\mathbf{y} = \mathbf{w} * \mathbf{x} = IDFT(DFT(\mathbf{w}) \cdot DFT(\mathbf{x})), \tag{2}$$

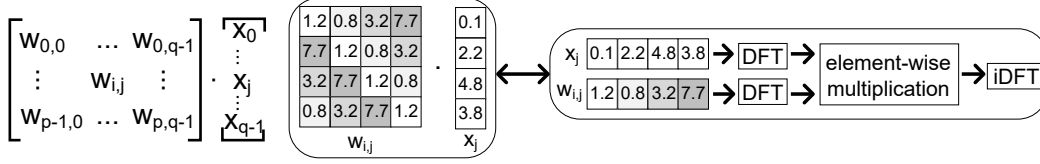

Figure 3: A block-circulant weight matrix.

where IDFT means inverse DFT. Frequency-domain convolutions greatly reduce inference latency of interactive HENNs, since unencrypted DFT and IDFT performed by the client introduce small computing overhead.

| Scheme | $MulPC$ | $AddCC$ | $rot$ | noise |
|--------|---------|---------|-------|-------|
| LoLa | $n_c$ | $n_c \cdot log_2 n_a$ | $n_c \cdot log_2 n_a$ | $\eta_0 \eta_m n_a + \eta_r (n_a\text{-}1)$ |
| LoLa+DFT | $3n'_c$ | $2n'_c \cdot log_2 n'_a$ | $2n'_c \cdot log_2 n'_a$ | $3\eta_0 \eta_m n'_a + 2\eta_r (n'_a\text{-}1)$ |

Table 1: The comparison of non-interactive HENN convolution schemes. The width, height and channels of convolution input, output: $I_w, I_h, I_c$ and $O_w, O_h, O_c$; The input channels, output channels, width and height of kernels: $I_c, O_c, f, f$. $N$ is ciphertext slots number. ($MulCP$: HE SIMD multiplications between plaintext and ciphertext; $AddCC$: HE SIMD additions between ciphertext and ciphertext; $rot$: HE rotation; $n_c = \lceil \frac{f^2 \cdot I_c \cdot I_w \cdot I_h}{N} \rceil$ and $n'_c = \lceil \frac{(I_c \cdot I_w \cdot I_h)^2}{N} \rceil$: ciphertext numbers of LoLa and LoLa+DFT; $n_a = I_c \cdot f^2$ and $n'_a = I_c \cdot I_w \cdot I_h$ : accumulation numbers of LoLa and LoLa+DFT, where $\lceil x \rceil$ is for roundup $x$ to the nearest integer. $\eta_0$: initial noise; $\eta_m$: MultPC noise; and $\eta_r$: rotation noise).

*Non-interactive HENNs.* Unlike interactive HENNs, non-interactive HENNs shown in Figure 1(b) have to use homomorphic DFT (HDFT) and IDFT (HIDFT), since the entire inference occurs on the server. Though a CKKS-based homomorphic DFT function with bootstrapping [11] exists, there is no BFV-based DFT or IDFT function. Even if we have BFV-based DFT and IDFT functions, the total HE number and noise of a non-interactive HENN will be greatly increased by HDFT and HIDFT that also require HE rotate-and-accumulate operations. DFT and IDFT can be summarized as

$$DFT(\mathbf{x})_t = \sum_{i=0}^{n-1} x_i \cdot \omega_n^{it}; IDFT(\mathbf{x})_t = \frac{1}{n} \sum_{i=0}^{n-1} x_i \cdot \omega_n^{-it} \tag{3}$$

where $\omega_n = e^{2\pi i/n}$. As Table 1 shows, if we naïvely apply HDFT and HIDFT on LoLa, LoLa+DFT increases more than $2\times$ rotations since $n'_a > n_a$ and $n'_c > n_c$, and thus introduces more noises. To maintain a larger noise budget for the noises, LoLa+DFT has to enlarge the HE encryption parameters, i.e., the ciphertext modulus $q$, and the polynomial degree $N$, which in turn prolong the latency of each $MulPC$, $AddCC$, and $rot$ operation.

## 2.4 Block-Circulant Weight Matrices

Recent works [13] compresses a weight filter of a plaintext NN into multiple blocks-circulant matrices to reduce inference computing overhead. The $n \times m$ weight matrix is divided into $p \times q$ square blocks, each of which ($\mathbf{w}_{i,j}$) contains $k \times k$ elements, where $0 \le i \le q - 1$ and $0 \le j \le p - 1$. We have $p = \frac{m}{k}$ and $q = \frac{n}{k}$. The $k \times k$ elements in a block can be derived by only $k$ elements with shift operations. The input vector is divided into $q$ parts, while the output vector is divided into $p$ parts. To compute each part of the output vector ($\mathbf{y}_i$) as shown in Figure 3, we can use

$$\mathbf{y}_i = \sum_{j=0}^{q} IDFT(DFT(\mathbf{w}_{i,j}) \cdot DFT(\mathbf{x}_j)) \tag{4}$$

The computational complexity of a FC layer is $\mathcal{O}(pqklogk)$.

## 2.5 Comparison against Prior Works

Table 2 shows the comparison of existing HENNs. A non-interactive HENNs, E2DM, is proposed to compute secure matrix multiplications using HE and MPC schemes. LoLa and CHET achieve the state-of-the-art performance in the non-interactive HENNs using HE ciphertext batching techniques. However, they still suffer from the long inference latency, due to massive and expensive homomorphic

| Features | E2DM [14] | LoLa [3] | CHET [15] | ENSEI [16] | MPCHE [8] | Ours |
|---|---|---|---|---|---|---|
| Non-interactive HENNs | ✗ | ✓ | ✓ | ✗ | ✗ | ✓ |
| Spectral Convolution & FC | ✗ | ✗ | ✗ | ✗ | ✗ | ✓ |
| Efficient homomorphic DFT | ✗ | ✗ | ✗ | ✗ | ✗ | ✓ |

Table 2: The comparison of HENNs.

rotations. ENSEI and MPCHE are proposed to reduce homomorphic rotations of interactive HENNs using spectral-domain convolutions. Nevertheless, their methods based on convolution theorem can not be applied to fully-connected operations. In addition, both ENSEI and MPCHE focus on interactive HENNs and cannot perform DFT and IDFT homomorphically. They enforce clients to process DFT and IDFT on the unencrypted data. Directly applying their methods on non-interactive HENNs prolongs the inference latency, due to the huge overhead of homomorphic DFT. Our Falcon includes an efficient homommorphic DFT technique and supports both spectral convolutions and FC operations, which significantly reduces the inference latency of non-interactive HENNs.

## 3 Falcon

### 3.1 BFV-based Homomorphic DFT

---
**Algorithm 1:** Homomorphic DFT (HDFT).

---
**Input:** a stacked input ciphertext $[x]$; the block size $k$
**Output:** a spectral ciphertext $[\hat{x}]$;
$[\hat{x}] = MultPC([x], \omega)$
**for** $i = 1; i <= log_2 k; i + +$ **do**
  $\quad [\hat{x}_i] = rot([\hat{x}], \frac{k}{2^i})$
  $\quad [\hat{x}] = AddCC([\hat{x}_i], [\hat{x}])$
**end**
**return** $[\hat{x}]$

---

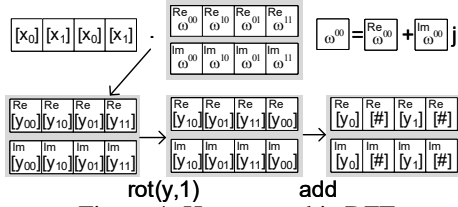
Figure 4: Homomorphic DFT.

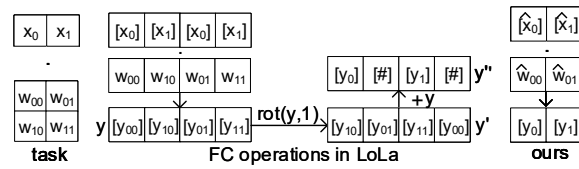
Figure 5: A Homomorphic Fully-Connected Layer.

To support homomorphic DFT shown in Equation 3 using BFV scheme, we should firstly quantize and encode the complex numbers ($\mathbf{w}_{i,j}$) of DFT conversion matrix.

*Quantization.* State-of-the-art non-interactive HENNs, e.g., LoLa, rely on the BFV protocol that supports only integer. The inputs, weights, activations, and outputs of the non-interactive HENNs are all integers. So $x_i$ of HDFT and HIDFT in Equation 3 is an integer. $\omega_n^{it}$ can be quantized as

$$Q(\omega_n^{it}) = \lceil S_1 \omega_n^{it} \rfloor = \lceil S_1 \cos(\frac{2\pi t}{n}) \rfloor - j \lceil S_1 \sin(\frac{2\pi t}{n}) \rfloor, \tag{5}$$

where $\lceil \cdot \rfloor$ is the rounding function converting a real number to an integer; and $S_1$ is an integer scaling factor. Through multiplying $S_1$, the real inputs, weights, and $\omega_n^{it}$ of a HENN are scaled to keep more digits after their decimal point during the rounding process. We quantized the inputs, activations, weights, and $\omega_n^{it}$ of a HENN with 8-bit. We integrated the quantized HENN into the forward propagation of the training to minimize the accuracy loss.

*Encoding Complex Numbers.* Unlike HEAAN [11], the BFV protocol cannot naturally support complex number. We encode the real part $(Re)$ $\lceil S_1 \cos(\frac{2\pi t}{n}) \rfloor$ and the imaginary part $(Im)$ $\lceil S_1 \sin(\frac{2\pi t}{n}) \rfloor$ of $Q(\omega_n^{it})$ in Equation 5 by two SIMD slots of a BFV ciphertext, so we have $C = Re + jIm$. For a HE complex addition $(C_0 + C_1)$, we have $(Re_0 + Re_1) + j(Im_0 + Im_1)$. For a HE complex multiplication $(C_0 \times C_1)$, we have $((Re_0 + Im_0)Re_1 - (Re_1 + Im_1)Im_0) + j((Re_0 + Im_0)Re_1 - (Re_1 + Im_1)Re_0)$.

*Homomorphic DFT and IDFT.* We present a BFV-based homomorphic DFT in Algorithm 1 to homomorphically convert an encrypted integer vector $[x]$ to its encrypted frequency-domain vector

$y=[\hat{x}]$. Here $\omega$ is the quantized DFT twiddle factors and its entries can be stacked and permuted in an offline phase. Table 3 shows a comparison of computational complexity between prior homomorphic DFT (FTHDFT) [17] and our Algorithm 1 when they both use the block circulant matrix technique. Algorithm 1 needs $(log_2 k)$ rotations and additions, and 1 $MultPC$ operations for each real-part vector $(Re)$ and imaginary-part vector $(Im)$ of input vector $x$. Figure 4 shows an example of our algorithm 1 with $k = 2$, and this example requires 1 multiplication, rotation and addition for each real-part vector $(Re)$ and imaginary-part vector $(Im)$.

| Scheme | #$MulPC$ | #$AddCC$ | #$rot$ | #Depth | noise |
|---|---|---|---|---|---|
| FHDFT | $3 \times log_2 k$ | $3 \times log_2 k$ | $2 \times log_2 k$ | $3 \times log_2 k$ | $\eta'^{3 \times log_2 k}$ |
| Ours | 1 | $2 \times log_2 k$ | $2 \times log_2 k$ | 1 | $\eta'$ |

Table 3: The comparison of homomorphic DFT schemes with $k$ points. Here $\eta' = \eta_0 \eta_m k + \eta_r (k-1)$.

## 3.2 Homomorphic Spectral FC Layer

Algorithm 2 shows our proposed homomorphic fully-connected (HFC) layer, which takes a spatial domain vector $x$ as the input, and outputs the encrypted results $[y] = [x] \cdot W$, where $W$ is the weight matrix at a FC layer. If a FC layer is not the first layer, $x$ already is packed into $l = N_c(\frac{Q}{k})$ ciphertexts. In this paper, we set $N_c(x) = \lceil \frac{I \cdot x}{N} \rceil$ be the ciphertext numbers to hold $x \times$ stacked inputs with size $I$, where $N$ is ciphertext slots number and $\lceil y \rceil$ is a function to roundup $y$ into the nearest integer. If $N$ is large enough, $x$ only requires one single ciphertext. Algorithm 2 then uses $HDFT$ to homomrophically convert each ciphertext $[x_i]$ to its spectral-domain value $[\hat{x}_i]$. Then a homomorphical element-wise multiplication between spectral input $[\hat{x}_i]$ and weight $\hat{W}_i$ is performed by one single $MultPC$ operation, where $\hat{W}_i$ can be pre-computed. According to Equation 4 and Figure 3, we do not need to accumulate the result entries inside a block, but we still need to accumulate the entries between $log_2 \frac{I}{k}$ blocks. Therefore, $log_2 \frac{I}{k}$ rotations and additions are required to accumulate these blocks. Then we perform an inverse $HDFT$ step to convert the spectral result $[\hat{v}_i]$ into its spatial value $[v_i]$. At last, we can use $AddCC$ to add the partial sums into one ciphertext. Table 4 compares the computational overheads and noise growth between LoLa and our Falcon. Falcon requires $2 log_2 \frac{I}{k} \cdot N_c(\frac{Q}{k})$ rotations, but LoLa needs $log_2 I \cdot N_c(O)$ rotations. Our experiments show that When $k >= 2$, Falcon costs less computations than LoLa. For noise growth, we set $\eta' = \eta_0 \eta_m I + \eta_r (I - 1)$. When $k > 1$, Falcon reduces the noise accumulation, thereby potentially enabling more efficient HE parameters. Figure 5 shows an example why our Algorithm 2 is better than our baseline LoLa. In this example, $k = 2$, $I = 2$, $O = 2$ and $N = 2$, LoLa requires $N_c(O) = 2$ ciphertexts and $2 \times log_2(I) = 2$ additions, rotations and multiplications. Our Falcon only requires $N_c(\frac{Q}{k}) = 1$ ciphertext and $1 \times log_2(\frac{I}{2}) = 0$ rotations. This is only a toy example and we should add the overhead of $HDFT$ into our method. In practice, when $I$, $O$ and $N$ are very large, the overhead of $HDFT$ is tiny compared to the other computations within a spectral FC, which is shown in section 5.

---

**Algorithm 2:** Homomorphic FC Layer (HFC).

**Input:** an input vector $x$ with size $I$, FC weights $W$ with size $I \cdot O$, the block size $k$;
**Output:** an output ciphertext $[y] = [x] \cdot W$;
$x$ is packed to $l = N_c(\frac{Q}{k})$ ciphertexts $\{[x_0], [x_1], ..., [x_{l-1}]\}$;
**for** $i = 0;\ i < l;\ i + +$ **do**
    $[\hat{x}_i] = HDFT([x_i])$;
    $[\hat{v}_i] = MultPC([\hat{x}_i], \hat{W}_i)$;
    **for** $j = log_2 \frac{I}{k};\ j > 0;\ j - -$ **do**
        $[\hat{y}_i] = rot([\hat{v}_i], j)$;
        $[\hat{v}_i] = AddCC([\hat{y}_i], [\hat{v}_i])$;
    **end**
    $[v_i] = inverse\ HDFT([\hat{v}_i])$;
    $[y_i] = AddCC([y_i], [v_i])$;
**end**
**return** $[y]$ from $[y_i]$ to $[y_{l-1}]$;

---

**Algorithm 3:** Homomorphic Convolutional Layer.

**Input:** an input tensor $x$ with size $I_c \cdot I_w \cdot I_h$, Kernels $W$ with size $I_c \cdot O_c \cdot f \cdot f$, block size $k$;
**Output:** the convolution result $[y]$;
$x$ is packed to $l = M_c(\frac{O'}{k})$ ciphertexts $\{[x_0], [x_1], ..., [x_{l-1}]\}$;
**for** $i = 0;\ i < l;\ i + +$ **do**
    $[\hat{x}_i] = HDFT([x_i])$;
    $[\hat{x}_i] = MultPC([\hat{x}_i], \hat{W}_i)$;
    **for** $j_c = 1;\ j_c <= log_2 I_c;\ j_c = j_c + k_c$ **do**
        $[\hat{x}_i] = rot([\hat{x}_i], \frac{I_w \cdot I_h \cdot I_c}{2^{j_c}})$;    $[\hat{x}_i] = AddCC([\hat{x}_i], [\hat{x}_i])$;
    **end**
    **for** $j_w = 1;\ j_w <= log_2 f;\ j_w = j_w + k_w$ **do**
        $[\hat{x}_i] = rot([\hat{x}_i], \frac{I_h \cdot f}{2^{j_w}})$;    $[\hat{x}_i] = AddCC([\hat{x}_i], [\hat{x}_i])$;
    **end**
    **for** $j_h = 1;\ j_h <= log_2 f;\ j_h = j_h + k_h$ **do**
        $[\hat{x}_i] = rot([\hat{x}_i], \frac{f}{2^{j_h}})$;    $[\hat{x}_i] = AddCC([\hat{x}_i], [\hat{x}_i])$;
    **end**
    $[y_i] = inverse\ HDFT([\hat{x}_i])$;
**end**
**return** $[y]$ from $[y_i]$ to $[y_{l-1}]$;

| Scheme | $\#MulPC$ | $\#AddCC$ | $\#rot$ | #Depth | noise |
|---|---|---|---|---|---|
| LoLa | $N_c(O)$ | $log_2 I \cdot N_c(O)$ | $log_2 I \cdot N_c(O)$ | 1 | $\eta'$ |
| Ours | $3N_c(\frac{O}{k})$ | $2log_2\frac{I}{k} \cdot N_c(\frac{O}{k})$ | $2log_2\frac{I}{k} \cdot N_c(\frac{O}{k})$ | 1 | $\frac{\eta'}{k}$ |

Table 4: The comparison of homomorphic FC operations with $I$ inputs and $O$ outputs.

### 3.3 Homomorphic Spectral Convolution Layer

Algorithm 3 shows how to perform a homomorphic spectral convolution. In this paper, $I_c$, $I_w$ and $I_h$ are the input channel number, input width and input height, respectively; kernel size and kernel output channel number are $f$ and $O_c$; The output channel number, height and width are $O_c$, $O_h$ and $O_w$. We set $M_c(x) = \lceil \frac{I_c \cdot I_h \cdot I_w \cdot x}{N} \rceil$ be the ciphertext numbers to hold $x\times$ stacked input. During a convolution, each input's sliding window with size of $I' = f^2 \cdot I_c$ will perform a dot-product operation. Each convolution needs $O' = \frac{O_c \cdot f^2}{s^2}$ dot-products, where $s$ is the stride size. Previous works use $M_c(O')$ ciphertexts to pack $O'\times$ stacked inputs, so that one homomorphic convolution operation is converted to $O'\times$ homomorphic dot-product operations on each sliding window with size $I'$. Table 5 shows that previous work requires $log_2 I' \cdot M_c(O')$ rotations, and noise growth is $\eta' = \eta_0\eta_m I' + \eta_r(I' - 1)$. By using the block-circulant matrix technique, we only need $M_c(\frac{O'}{k})$ ciphertexts as shown in Algorithim 3, which potentially reduces $k\times$ homomorphic rotations, multiplications and additions. In addition, the dot-product of each ciphertext in Algorithm 3 only needs to accumulate entries between blocks, which is implemented by setting $k_c \cdot k_w \cdot k_h = k$. Thus, each dot-product with size $I'$ only requires $\frac{I'}{k}$ additions and rotations to accumulate its partial results. Table 5 concludes the computational overheads and noise growth of LoLa and our Falcon. Falcon reduces $\sim \frac{k \cdot log_2(k)}{2}$ rotations, additions and multiplications over LoLa. Flacon has $k\times$ less noise increase than LoLa.

| Scheme | $\#MulPC$ | $\#AddCC$ | $\#rot$ | #Depth | noise |
|---|---|---|---|---|---|
| LoLa | $M_c(O')$ | $log_2 I' \cdot M_c(O')$ | $log_2 I' \cdot M_c(O')$ | 1 | $\eta'$ |
| Ours | $3M_c(\frac{O'}{k})$ | $2log_2\frac{I'}{k} \cdot M_c(\frac{O'}{k})$ | $2log_2\frac{I'}{k} \cdot M_c(\frac{O'}{k})$ | 1 | $\frac{\eta'}{k}$ |

Table 5: The comparison of homomorphic convolution schemes.

## 4 Experimental Methodology

**Dataset, Networks and Operation Details.** Our datasets include MNIST [18] and CIFAR-10 [19]. The network architecture for MNIST dataset is same to LoLa [20], but we replace the spatial-domain middle layers into frequency domain. The network architecture and operations are summarized in Table 6. The block size $k$ of circulant matrix is set as 8 so that accuracy is not decreased. We evaluated a 3-layer CNN same to LoLa [20] on CIFAR-10. To keep original accuracy, the block size $k = 16$ of circulant matrix is used.

**Cryptosystems Settings.** We use BFV scheme in SEAL [21] to implement Falcon. For MNIST and CIFAR-10, the plaintext modulus $t = 2148728833 \times 2148794369 \times 2149810177$, modulus degree $N = 16384$, coefficient modulus $Q =\sim 440$ bits. More specific encryption parameters settings are shown in Table 7 and Table 8. The security level is larger than 128 bits which is verified by $lwe\_estimator$ [22]. To have fair comparisons with baselines, We ran all experiments on the same Azure standard B8ms virtual machine with 8 vCPUs and 32GB DRAM.

## 5 Results and Analysis

We compared Falcon against the state-of-the-art works including CryptoNets [23], Faster CryptoNets (FCryptoNets) [24], nGraph-HE [25], CHET [15] and LoLa [20]. The homomorphic operation numbers (HOPs), encryption parameter bits, message size, latency and accuracy are summarized in Table 7. HOPs is the sum of all homomorphic operation number including $MultPC$, $MultCC$, $AddCC$ and $rot$. Message size is the size of encrypted input and output that the client needs to transmit.

### 5.1 MNIST

As Table 7 shows, CryptoNets runs one MNIST image inference in 205 seconds, since it uses an inefficient encoding method. Specifically, CryptoNets encodes a pixel into a single message, therefore a $28 \times 28$ MNIST image is encoded and encrypted into 784 ciphertexts. nGraph-HE reduces 34% inference latency of CryptoNets by multiple compiler optimizations. FCryptoNets makes use of the

| Layer | Input size | Representation | Falcon HE operation | #Rot |
|---|---|---|---|---|
| convolution layer | $25 \times 169$ | convolution | convolution vector - row major multiplication | 0 |
| ($f^2$=25, $O_c$=5) | $5 \times 169$ | dense | combine to one vector using 4 rotations and additions | 4 |
| square layer | $1 \times 845$ | dense | square | 0 |
| DFT | $1 \times 845$ | dense | homomorphic DFT | 12 |
| FC layer | $1 \times 845$ | dense | stack vectors using $2log_2\frac{Q}{k}$ rotations and additions | 8 |
| ($I$=845, $O$=100) | $1 \times (845 \times \frac{Q}{k})$ | stacked | homomorphic FC | 14 |
| Inverse DFT | $1 \times 100$ | interleave | inverse homomorphic DFT | 12 |
| square layer | $1 \times 100$ | interleave | square | 0 |
| FC layer ($O$=100, $O$=10) | $1 \times 100$ | interleave | interleaved vector - row major multiplication | 70 |
| output | $1 \times 10$ | sparse | | |

Table 6: Falcon message representations and operations on MNIST. Falcon only replaces the first FC layer by HSFC. Falcon only uses (12+8+14+12)=46 rotations, reducing **70%** rotation numbers in the first FC layer compared to LoLa which consumes 151 rotations.

weights sparsity in neural networks to speedup the MNIST inference into 39.1 seconds. However, due to the inefficient encoding method, they still suffer from huge HOPs ($> 67K$) and big message size ($> 368MB$). LoLa is proposed to efficiently encode one image's multiple pixels into a single message so that one ciphertext is able to contain all 784 MNIST pixels. In practice, LoLa encodes $f^2$=25 messages, each of them contains $O_w \cdot O_h$=169 pixels, to fast process first-layer convolution operations. Although EVA [26] and LoLa significantly reduce HOPs and inference latency, they introduce 2K and 225 expensive rotations, respectively. Rotations are the computational bottleneck of LoLa. LoLa directly uses the over-pessimistic encryption parameters generated in SEAL [21], e.g. 440-bit $Q$ and 14-bit $N$. According to the noise analysis in Table 4 and Table 5, {40, 60, 80, 60, 70} noise bits are accumulated by convolution, square, FC, square, FC layers respectively. So 340-bit $Q$ and 14-bit $N$ are required to guarantee a 128-bit security level. Using 340-bit $Q$, LoLa' is able to reduce 0.1-second latency of LoLa. Compared to LoLa', our work Falcon reduces ~10-bit noise bits using block-circulant matrix, but introduces 100-bit noise bits because of DFT and inverse DFT, thereby Falcon consumes 430-bit $Q$. Our work Falcon reduces 56% rotation numbers and achieves the MNIST inference latency in ~ 1 seconds without accuracy decrease as shown in Table 7.

Table 6 reports the batching representations and operations that Falcon applies in each layer. These batching representations are defined in LoLa. Since the second FC layer of LoLa occupies ~ 63% of the total latency, we replace this layer by our Homomorphic FC operations and keep the other layers same to LoLa. Falcon performs a convolution vector - row major multiplication in the first layer without rotations, generating 5 dense ciphertexts. These 5 dense ciphertexts can be combined into one ciphertext by 4 rotations and additions. This ciphertext is squared using a single multiplication, outputting a dense ciphertext, which is homomorphically converted into a frequency-domain ciphertext by our HDFT algorithm shown in Algorithm 1.Then 13 copies of the spectral ciphertext are stacked before applying the Homomorphic FC layer, which is done using 16 rotations when we support complex-number encoding. Then only 1 round HSFC is performed to generate $1\times$ 100 interleaved ciphertext. The interleaved ciphertext is then squared by one single multiplication. The last operations is an interleaved vector - row major multiplication, which requires 70 rotations. Falcon only uses 46 rotations, reducing 70% rotation numbers in the first FC layer compared to LoLa which consumes 151 rotations. Falcon needs 120 rotations in total for MNIST inference, reducing 47% rotations compared to the 225 rotations in the LoLa, thereby Falcon reduces 45% latency.

| Scheme | HOPs | $AddCC$ | $MultPC$ | $MultCC$ | $Rot$ | $N$(bits) | $Q$(bits) | Message Size | Latency(s) | Acc(%) |
|---|---|---|---|---|---|---|---|---|---|---|
| CryptoNets | 612K | 312K | 296K | 945 | 0 | - | - | 368MB | 205 | 98.95 |
| nGraph-HE | 612K | 312K | 296K | 945 | 0 | 14 | 350 | - | 135 | 98.95 |
| FCryptoNets | 67K | 38K | 24K | 945 | 0 | 13 | 219 | 411MB | 39.1 | 98.71 |
| EVA | 10K | 4K | 4K | 3 | 2K | 15 | 480 | 63MB | 121.5 | 99.05 |
| LoLa | 798 | 393 | 178 | 2 | 225 | 14 | 440 | 51MB | 2.2 | 98.95 |
| LoLa' | 798 | 393 | 178 | 2 | 225 | 14 | 340 | 51MB | 2.1 | 98.95 |
| **Falcon** | 626 | 312 | 204 | 2 | 120 | 14 | 430 | 51MB | **1.2** | 98.95 |

Table 7: The MNIST results.

## 5.2 CIFAR-10

Table 8 shows the inference latency, accuracy and HOPs of existing works on CIFAR-10. nGraph-HE implements CIFAR-10 inference in 1628 seconds with 62.1% accuracy. FCryptoNets improves 14.6% accuracy compared to nGraph-HE with a deeper and more complex neural network architecture, but it suffers from huge latency with 39K seconds. EVA and LoLa support same vector representations

and operations, but they are based on HEAAN and BFV respectively and encryption parameters, thereby they have different latency. LoLa introduces 53K expensive rotations, each of rotations has $\sim 10\times$ latency of multiplication $MultPC$. Our work Falcon removes 86% rotations and reduces the latency from 730 seconds to 107 seconds. Table 9 reports the batching representations and operations that Falcon applies in each layer. Since the second FC layer of LoLa occupies $\sim 97\%$ of the total latency, we replace this layer by our homomorphic convolution and keep the other layers same to LoLa. Falcon uses 7674 rotations, reducing 86% rotation numbers in the second convolution layer compared to LoLa which consumes 52975 rotations. Our baseline LoLa can be improved by LoLa' using more proper encryption parameter $Q$.

| Scheme | HOPs | $AddCC$ | $MultPC$ | $MultCC$ | $Rot$ | $N$(bits) | $Q$(bits) | Message Size | Latency(s) | Acc(%) |
|---|---|---|---|---|---|---|---|---|---|---|
| EVA | 150K | 67K | 67K | 9 | 16K | 16 | 1225 | 63MB | 3062 | 81.5 |
| LoLa | 123K | 61K | 8.2K | 2 | 53K | 14 | 440 | 210MB | 730 | 76.5 |
| LoLa' | 123K | 61K | 8.2K | 2 | 53K | 14 | 330 | 210MB | 730 | 76.5 |
| **Falcon** | 21K | 10k | 11.9K | 2 | 7.9K | 14 | 430 | 210MB | **107** | **76.5** |

Table 8: The CIFAR-10 results.

| Layer | Input size | Representation | Falcon homomorphic operation | $\#Rot$ |
|---|---|---|---|---|
| convolution | $64 \times 196$ | convolution | convolution vector - row major multiplication | 0 |
| ($f^2$=64, $O_c$=83) | $83 \times 196$ | dense | combine to one vector using 82 rotations and additions | 82 |
| square | $1 \times 16268$ | dense | square | 0 |
| DFT | $1 \times 16268$ | dense | homomorphic DFT | 12 |
| convolution | $1\times 16268$ | dense | Homomorphic convolution | 7650 |
| ($f^2$=25, $O_c$=163) | $\frac{4075}{k} \times k$ | sparse | combine to one vector using $\frac{4075}{k}$-1 additions | 0 |
| inverse DFT | $1 \times 4075$ | dense | homomorphic DFT | 12 |
| square | $1 \times 4075$ | dense | square | 0 |
| FC ($I$=4075, $O$=10) | $1 \times 4075$ | dense | 10 dense vector-row major multiplication | 120 |
| output | $1 \times 10$ | sparse | | |

Table 9: Falcon operations on CIFAR-10. Falcon replaces the second convolution layer by HSConv. Falcon only uses (12+7650+12)=7674 rotations, reducing **86%** rotation numbers in the second convolution layer compared to LoLa which consumes 52975 rotations.

# 6  Conclusion

In this paper, we propose Falcon, a low-latency deep neural network on encrypted data, which consists of a homomorphic DFT unit, a Homomorphic FC unit and a Homomorphic convolution unit based on block-circulant matrix. Our experimental results show Falcon reduces the inference latency by $45.45\% \sim 85.34\%$ over prior HENNs on various datasets. Falcon is the first frequency-domain non-interactive HENNs.

## Broader Impact

Falcon enables a low-latency privacy-preserving neural network inference on encrypted data. With Falcon, users can enjoy low-latency secure inference services. In particular, users are able to receive low-latency and powerful machine learning inference services by uploading their sensitive data without concerning data privacy. Falcon has no negative impact on our society. If our proposed method fails, the latency of secure inferences will be prolonged.

## Acknowledges

The authors would like to thank the anonymous reviewers for their valuable comments and helpful suggestions. This work was partially supported by the National Science Foundation (NSF) through awards CCF-1908992 and CCF-1909509.

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
