[Reviews · NeurIPS 2020]

Review 1

Summary and Contributions: The paper describes a new optimization for speeding up the homomorphic evaluation of linear layers in neural network in the context of privacy-preserving neural network inference. The authors replace the rotate and sum paradigm for computing convolutions and FC layers with an FFT based. To simplify the implementation they rely on block-circulant CNNs. On the whole they show a 45-85% reduction in inference latency compared to state of art

Strengths: The main strength of the work is the empirical evaluation of an FFT based approach to implementing the linear layers in a neural network. Rather than directly using rotations to implement a convolution the authors propose implementing rotations for the FFT and IFFT layers. They show 2-4x reduction in latency doing the same.

Weaknesses: There are a couple of concerns with the approach as shown in the paper. The first concern seems to be that a fairly weak baseline was assumed. For example, the RHECNN baseline in Fig. 4 results in ciphertexts that are not packed. However prior art such as Gazelle already shows how to generate packed ciphertexts with fewer rotations. The second weakness is the activations functions chosen for evaluation. This work assumes square activations which are easy to implement with HE but are known to result in sub-optimal training for real world problems. Use of more conventional non-linearities like ReLU results in significantly increased runtime for the non-linear layers and as such reducing the efficacy of the any runtime improvements in the linear layers -- Post Rebuttal: I do not agree with the authors rebuttal about the convolution routine using a competitive baseline. As presented in Fig 4a, the baseline that authors consider starts with packed ciphertexts but ends up with unpacked intermediates. This is supported by the authors explanation of needing c_o * w_o * h_o * log_2 (c_i * f^2 ) rotations as per Section 3.3. Prior art e.g. Gazelle (Table 4, Row 2) shows how to accomplish this very operation using exactly one hoisting operation followed by c_i * f^2 permutations. For completeness the setting of this paper is the same c_i = c_n (The authors mention in Section 3.3 that they are able to concatenate all c_i channels into a single length-N ciphertext). I am thus not convinced about the authors explanation of the baselines they have used.

Correctness: The technical results mentioned in the paper are correct in the context of the baseline used by the authors. (As mentioned above the use of a stronger baseline will change the relative latency improvement

Clarity: The paper is well written. Fig 4. could benefit from a more clear presentation. It is a bit hard to understand how the ciphertexts are organized from the figure alone.

Relation to Prior Work: The authors clearly highlight the differences in their approach when compared with the LoLa baseline. However stronger baselines from the MPC+HE literature could be considered to complete the picture.

Reproducibility: Yes

Additional Feedback:


Review 2

Summary and Contributions: The paper introduces an improvement to the linear layers in neural network inference (convolution and fully connected) using homomorphic encryption. This is done via the Fast Fourier Transform.

Strengths: The material is introduced in some depth and supported by benchmarks.

Weaknesses: The authors fail to compare their approach to Jian et al. (CCS '18) who use the inherent algebraic properties to compute matrix multiplications. Similarly, they claim that Costache et al. heavily use ciphertext rotations whereas rotations aren't even mentioned in said work. Post rebuttal: I concede that later works are a better baseline, and I appreciate the proposed additions and corrections.

Correctness: The claims are credible.

Clarity: Grammar and spelling could be improved (see below), and it seems to make heavy use of negative space around figures and converting lists to running text in Section 2 in order to reduce the length. I think that the technical introduction is somewhat confusing, and I'm not convinced that heavily relying on a graphical representation of ciphertext operations is appropriate in this context. - It is confusing that "log" is written in italic in formulas. - p3: "in (the) Brakerski/Fan-Vercauteren scheme" - p3: Singel Instruction Multiple Data - p4: in time-domain - p5: 163 homomorphic dot product(s) - p5: we derive c_0 intermediate ciphertext [t] who - p6: repeat (the) previous process - p6: generating output with - p6: by traditional s-stride convolution method - p7: Flacon - p7: cihertex

Relation to Prior Work: The improvement to cited work is discussed well.

Reproducibility: Yes

Additional Feedback:


Review 3

Summary and Contributions: The paper introduces a new world-record in privacy preserving inference. The authors introduce a new way to speed-up the evaluation of a neural-network on data encrypted with Homomorphic Encryptions. They make a nice observation that the latency of current state-of-the-art methods is attributed mainly to the rotation operation when doing matrix-vector operation. Therefore, they suggest doing the multiplication in the frequency domain such that less rotations are needed. This allows them to improve the latency by 45%-85%.

Strengths: The problem is important, and the authors break the state-of-the-art in the relevant benchmarks

Weaknesses: The paper is not easy to follow and the novelty with respect to previous work is not clear

Correctness: The claims seems correct and the empirical evaluation seems adequate.

Clarity: The presentation is hard to follow and confusing at places. In several places in the paper (for example, line 150, line 186) the authors refer to the ReLU activation function. How do you apply ReLU with HE? In previous work (LoLa, CryptoNets, …) ReLU was avoided and the square activation was used instead. However, here you claim that ReLU are used and use it to justify the scaling argument in line 186. Please explain. It is also hard to follow the main algorithm – even the figures are confusing.

Relation to Prior Work: The main idea in this work is to reduce the number of rotation operation in matrix-vector multiplications using FFT. The FFT is performed by multiplying by the Hadamard matrix. How does this compare with methods such as the one described in Section 3.1 of Jiang et al, “Secure Outsourced Matrix Computation and Application to Neural Networks”, 2018 or the method described in Bian et al “ENSEI: Efficient Secure Inference via Frequency-Domain Homomorphic Convolution for Privacy-Preserving Visual Recognition”, 2020 (the last one uses secret shares as a part of the protocol but the idea of using FFT is present).

Reproducibility: Yes

Additional Feedback: 1. Line 38: “are proposed” – this should be rephrased to something like “were proposed” or “proposed” 2. Line 150: the term HReLU is confusing since square activation is used an not a ReLU. I would suggest using different notation such as HSquare or HActivation 3. Line 167: CryptoNets do not use either B/FV or HEAAN. Moreover, the LoLa paper shows that the latency saving cannot be attributed to only the encryption scheme 4. Line 186: How do 5. Bibliography uses inconsistent formats. Some items cotain the full author list while other use et al. notation. While using et al in the body of the work is common, using it in the bibliography should be avoided 6. References [2] and [10] refer to the same paper ==== after rebuttal ==== The paper can enjoy from adding few details: 1) for the cifar-wide networks: the details of these networks are not clear (the topology of the network). I think the reference should go to the paper https://arxiv.org/abs/1602.07360 if I understand correctly 2) for tables 4 & 5: were all experiments done using the same hardware? it seems that some of the numbers use timing from previous papers that may have been using different hardware and therefore the comparison is not clear

[Author Response · NeurIPS 2020]

We thank the reviewers for their careful reading of the manuscript and their constructive suggestions.

**Reviewer-1, Weak baseline (baseline ciphertext is not packed), comparison against MPC+HE baselines**: First, our RHECNNS baselines, LoLa [7] and CHET [9], use the ciphertext packing technique shown in Fig.4. LoLa [ICML'19] and CHET [PLDI'19] are the state-of-the-art HE-enabled neural networks. Compared to secure Multi-Party Computation (MPC), HE supports non-interactive operations and greatly reduces the communication cost. Gazelle [13] is described at lines 73 - 75 of our paper. Gazelle is based on interactive MPC+HE, so it suffers from huge communication overhead between clients and server (1.2 GB per CIFAR-10 image). The MPC protocol requires clients to be always online to communicate with servers. In the setting of Machine Learning as a Service, it is difficult for some clients to stay online and share the data by a stable and high speed connection during the entire service. Our Falcon is proposed to reduce the non-interactive HE operations, NOT the MPC+HE hybrid operations.

**Reviewer-1, Why not ReLU activation?**: (1) Our baselines LoLa [7] and CHET [9] show the square polynomial approximation have competitive accuracy, compared to the original ReLU on MINIST and CIFAR-10. (2) We use the SAME activation approximation as LoLa [7] and CHET [9] for a fair comparison.

**Reviewer-2/3, Novelty (Comparison against E2DM [CCS'18])**: E2DM [Jiang et.al at CCS'18] is not the state-of-the-art secure inference. One of our baselines CHET [9] claims it has better performance than prior works like E2DM. We would like to point out that we compared our work against state-of-the-art RHECNNs, including CHET and LoLa that obtain better performance than E2DM on CIFAR-10. E2DM is shown effective in matrix multiplications, not in the modern convolution operations which are well-studied by CHET [7] and LoLa [9]. In addition, E2DM is mainly proposed to reduce the number of multiplications, but our Falcon is proposed to reduce the number of rotations that are much more expensive. We will add E2DM into related work and compare it against our Falcon in the revised version.

**Reviewer-2, Rotations in Costache's paper [23]**: We agree that Costache's paper [23] has not heavy rotations. Instead of [23], homomorphic DFT [25] heavily depends on rotation. In our Falcon, [25] works as the baseline since it is the state-of-the-art homomorphic DFT and outperforms the Costache's work [23]. We show that our Falcon outperforms the state-of-the-art homomorphic DFT [25] in section 3.

**Reviewer-3, Novelty (Difference with ENSEI (Bian Song, et al. CVPR'20))**: Firstly, we would like to point out that ENSEI and our work Falcon target on different cryptography protocols. ENSEI adapts interactive HE+MPC setting, but Falcon uses non-interactive HE setting. We described the fact that MPC-based ENSEI suffers from high communication overhead in line 73 - 75. **Simply adopting ENSEI in non-interactive HE-based networks is NOT trivial**. This is because cheap DFT and IDFT in plaintext domain CANNOT be performed by clients in the non-interactive HE setting, instead expensive homomorphic DFT and IDFT are required to be performed by servers. As Figure 1 shows, LoLa-S (adopting ENSEI into LoLa) simply using DFT on non-interactive HE setting prolongs the baseline LoLa's latency because of the expensive and essential homomorphic DFT and IDFT operations. Our work Falcon proposes efficient homomorphic DFT in the algorithm 1 to solve the above problem. Second, ENSEI is only shown effective in the convolutional layers, not the fully-connected layers. This is because dot-product operations in the fully connected layers CANNOT be directly applied into convolution theorem. In contrast, Falcon uses block-circulant matrix to support underlying dot-product operations, so both convolution and fully-connected operations are supported well in our work. We will highlight that LoLa-S in figure 1 refers to the spectral-version LoLa using the similar method in ENSEI. Falcon's novelty and contributions can be concluded by three points: 1. We propose efficient homomorphic DFT and IDFT algorithms. 2. We use block-circulant matrix to support efficient spectral convolution and fully-connected operations in encrypted data. 3. Our experiments show that Falcon can be applied into any non-interactive HE networks for reducing expensive HE operations.

**Reviewer-2/3, Typos and References formats**: Thanks for reviewers' correction. We will fix them in the revised version. Especially thanks for the advice of reviewer 3 on "Changing the name $HReLU$ into $HSquare$ or $HActivation$."

[Meta-Review · NeurIPS 2020]

Overall the reviewers found this paper interesting and novel and expect it to inspire future work in the field.